# Influence of Lignin and Polymeric Diphenylmethane Diisocyante Addition on the Properties of Poly(butylene succinate)/Wood Flour Composite

**DOI:** 10.3390/polym11071161

**Published:** 2019-07-08

**Authors:** Chan-Woo Park, Won-Jae Youe, Song-Yi Han, Ji-Soo Park, Eun-Ah Lee, Jung-Yoon Park, Gu-Joong Kwon, Seok-Ju Kim, Seung-Hwan Lee

**Affiliations:** 1College of Forest & Environmental Science, Kangwon National University, Chuncheon 24341, Korea; 2Forest Products Department, National Institute of Forest Science, Seoul 02455, Korea; 3Kangwon Institute of Inclusive Technology, Kangwon National University, Chuncheon 24341, Korea

**Keywords:** wood-plastic composite, twin-screw extrusion, kraft lignin, pMDI

## Abstract

Poly(butylene succinate) (PBS)/wood flour (WF) composites with different WF content were prepared by twin-screw extrusion at 160 °C. With increasing WF content, the tensile strength of the PBS/WF composite without polymeric diphenylmethane diisocyante (pMDI) decreased, while that of the composite with pMDI increased. The addition of kraft lignin (KL) deteriorated the tensile properties of the composites both with and without pMDI. The melt flow index (MFI) decreased with increasing WF content, but increased with increasing KL content. The addition of pMDI caused an increase in the melt viscosity of the PBS/WF and PBS/WF/KL composites, resulting in a decrease in the MFI. The composites had lower thermal stability than neat PBS. The exotherms of the PBS/WF (50/50) composite appeared at a higher temperature than that of the neat PBS, but the PBS/WF/KL (50/50/20) composites had similar exotherms as the neat PBS. The addition of KL caused a decrease in the crystallization rate of PBS.

## 1. Introduction

Wood-plastic composite (WPC) materials are constructed of a thermoplastic polymer and wood flour (WF), and are produced by mixing the molten thermoplastic resin and WF through extruding or kneading [1,2,3,4]. WPC is considered an eco-friendly material. Waste wood and recycled plastic can be used as raw materials for WPC production, and WPC can also be recycled [3,4,5,6,7]. WPC can overcome disadvantages of wood materials such as low processability, vulnerability to damage by blight and harmful insects, and sensitivity to moisture, and can represent the texture of the wood [8,9]. Owing to these advantages, WPC has been widely applied as a building material and interior material [6,10]. Polyethylene (PE), polypropylene (PP), and polyvinyl chloride (PVC) are most commonly used for WPC production [1,7]. However, recently, microplastics, which are particles smaller than 5 mm in diameter derived from non-biodegradable polymers, have contributed to water pollution and threatened human health. For that reason, biodegradable polymers, such as poly(lactic acid) (PLA), poly(butylene succinate) (PBS), and polycaprolactone (PCL) are considered important materials for the development of eco-friendly composites [2,11,12,13].

PBS is a promising biodegradable polymer produced through the polycondensation of 1,4-butanediol with succinic acid [14,15,16]. Because PBS has outstanding properties including biodegradability, biocompatibility, flexibility, and high processability and strength, as well as excellent thermal and chemical resistance, it is expected that PBS can be used to produce WPC with excellent properties [16,17,18,19,20,21,22]. The WPC produced from PBS and WF has high environmental value as a biodegradable composite. However, as WF has no melting point, a high content of WF in the WPC can cause a decrease in its melt flowability, which results in deterioration of the workability and quality of the WPC [23,24,25,26]. The melt flowability of WPC can be increased simply by adding lignin isolated from the wood. Lignin is the most abundant aromatic biopolymer, accounting for 15%–40% of the lignocellulose [27,28]. Because lignin has a softening point in the range of 130–180 °C and is compatible with WF, lignin can be applied as an additive to improve the melt flowability of the WFC. In a previous study [13], it was demonstrated that the addition of kraft lignin (KL) to PCL enhanced the melt flow index (MFI), indicating a lower melt viscosity (MV). The lignin addition not only can overcome some disadvantages of the WPC but can also increase the lignocellulosic biomass content in the WPC. However, the lignin addition led to a reduction in the tensile strength of the composite because the lignin is incompatible with most thermoplastic polymers. This deterioration in the tensile strength of PCL/KL composites due to the presence of KL has been overcome by adding polymeric diphenylmethane diisocyanate (pMDI) as a coupling agent. The addition of pMDI can effectively improve the interfacial adhesion between the lignocellulose or lignin and aliphatic polyesters such as PCL, PBS, polyhydroxyalkanoate (PHA), and polyhydroxybutyrate (PHB) [13,29,30,31].

In this study, PBS/WF composites with different ratios of PBS/WF were prepared through twin-screw extrusion at 160 °C with a screw speed of 55 rpm. The effect of addition of KL and pMDI on the physicomechanical and thermal properties of the PBS/WF composites was investigated.

## 2. Materials and Methods

PBS pellets (Solpol-5000) were purchased from Gio Soltech Co. Ltd. (Wonju, Korea). Commercial WF (Lignocel® C120) was purchased from J. Retenmaier & Sohne Co. (Rosenberg, Germany). The pMDI used as a coupling agent was obtained from Kumho Mitsui Chemicals Co. (Seoul, Korea). The black liquor for lignin isolation was produced from hardwood chips by Moorim Pulp & Paper Co. (Ulsan, Korea). KL was isolated from the black liquor [28]. After adjusting the pH of the black liquor to ~2.0 using a concentrated hydrochloric acid solution, the precipitated lignin was recovered by filtration. This precipitate was repeatedly washed with deionized water, and then dried at 60 °C in an oven for 2 weeks.

The PBS, WF, and KL were vacuum-dried at 40 °C and –98 kPa for 48 h. The PBS and WF were pre-mixed at ratios of 90/10, 70/30, and 50/50. KL corresponding to 5, 10, and 20 phr on the basis of mixture weight was added to the PBS/WF (50/50) mixture. To investigate the effect of addition of a coupling agent, 2 phr pMDI was added to each mixture. The material formulation was indicated in Table 1. The mixtures were placed in a twin-screw extruder (BA-11, Bautek Co., Pocheon, Korea) with a length-to-diameter (L/D) ratio of 40. The WPC composites were prepared by twin-screw extrusion at 160 °C with a screw speed of 55 rpm.

The morphologies were observed using scanning electron microscope (SEM; S-4800, Hitachi, Ltd., Tokyo, Japan) at the Central Laboratory of Kangwon National University. For the morphology observation, the fractured WPC composites were coated with iridium using a high-vacuum sputter coater (EM ACE600, Leica Microsystems, Ltd., Wetzlar, Germany). The coating thickness was approximately 2 nm.

The tensile properties of the composites were investigated according to the American Society for Testing and Materials (ASTM) D638 standard. The WPC composites were hot-pressed at 160 °C for 1 min to form sheets. The specimens were cut with a Type V sample cutter, as described by the ASTM D638 standard, and maintained in a thermo-hygrostat at a relative humidity of 65% to minimize the effects of relative humidity on the tensile properties. The tensile tests were conducted using a universal testing machine (H50K, Hounsfield Test Equipment, Redhill, UK) at a cross-head speed of 10 mm/min. A minimum of eight specimens of each sample were tested and the average values were reported.

The MFI and MV of the WPC composites were measured with a melt flow indexer (MFI 4050, Rhopoint Instruments, Ltd., Hastings, UK). The samples were pre-heated in a vessel for 2 min at 180 °C, and then the MFI and MV were measured using a die with a diameter of 2.09 mm and load cells of 2.16, 5.00, 7.50, 10.0, and 12.5 kg. The measurement distance was set at 25.4 mm, and the MFI and MV were calculated automatically by the software for the melt flow indexer.

Thermogravimetric analysis (TGA) of the WPC composites was conducted using a thermogravimetric analyzer (Q500, TA instruments Inc., New Castle, DE, USA) at the Central Laboratory of Kangwon National University. The samples (15–20 mg) were heated on a platinum pan under a nitrogen atmosphere. The range of scanning temperatures was 25–500 °C, with a heating rate of 10 °C/min. Derivative thermogravimetry (DTG) was analyzed by measuring the mass loss with respect to time.

Differential scanning calorimetry (DSC) measurements were performed with a differential scanning calorimeter (Q2000, TA instrument, DE, USA.). The samples were first heated to 150 °C at a heating rate of 20 °C/min, and then held for 5 min to eliminate the thermal history. To examine the non-isothermal crystallization kinetics, the samples were cooled to 30 °C at rates of 10, 20, 30, and 40 °C/min and then reheated to 150 °C at 20 °C/min.

## 3. Results and Discussion

### 3.1. Morphological Characteristics

Figure 1 indicates the morphological characteristics of WL and KL. The WF had the rod-like morphology with micron-scale and showed the cellulose microfibril bundles at high magnification. The KL indicated that nanoscale lignin particles came together to form micron-scale particle aggregates.

Figure 2 shows the effects of the WF content and pMDI addition on the morphological characteristics of the fractured surface of the PBS/WF composites. With increasing WF, the fractured surface of the composite became rougher. In the PBS/WF (50/50) samples, a number of cavities with sizes of 5–10 μm were observed, caused by the removal of WF particles during fracture. These features indicate a weak interfacial adhesion between the PBS matrix and the WF. Most of the aliphatic polyester polymers are hydrophobic, resulting in incompatibility with the hydrophilic reinforcing filler, particularly the natural fiber containing hydrophilic polysaccharides. The pMDI was used as a coupling agent to increase the interfacial adhesion between the PBS matrix and the WF. The PBS/WF composites with 2 phr pMDI had a smoother fractured surface than those without pMDI. In addition, the number of cavities in the fractured surface was reduced. These results indicate that the coupling agent can improve the compatibility between the PBS and the WF.

KL at 5, 10, and 20 phr on the basis of the weight of the composite was added to the PBS/WF (50/50) composites. The effect of KL addition on the morphological characteristics of the PBS/WF (50/50) composites with and without pMDI (2 phr) is shown in Figure 3. KL particles, which typically have sizes ranging from a few nanometers to micrometers, were observed in all samples. As the KL content increased, the size of the aggregated lignin particles and the pores formed by the removal of KL and WF particles in the composites increased. This indicates that the hydrophobic KL was also incompatible with the PBS matrix. The compatibility of the KL with the aliphatic polyesters can be improved by adding pMDI. In a previous study [13], our research team investigated the effect of pMDI on the properties of KL/PCL composites. With the addition of pMDI to the KL/PCL composite, the fractured morphology became smooth and clean, resulting from the improvement in the interfacial adhesion between the two polymers. Figure 3 also shows the influence of pMDI on the compatibility between the KL and the PBS matrix. With the addition of 2 phr pMDI to the PBS/WF composites containing 5, 10, and 20 phr KL, the morphology of the fractured surface of the composites became smoother than that of the composites without pMDI. In the samples with added pMDI, the composites with 5 and 10 phr KL exhibited cleaner morphological characteristics compared to the samples with no KL, and appeared to be filled with the KL. The pMDI improved not only the interfacial adhesion between the WF and PBS, but also the compatibility with the KL. The PBS/WF/KL (50/50/20) composite with 2 phr pMDI exhibited aggregated KL particles and cavities, indicating a rough fractured surface due to an excessive KL content.

### 3.2. Tensile Properties

Figure 4 shows the effect of the WF content on the tensile strength, elastic modulus, and elongation at break of the PBS/WF composites with and without pMDI. The composites without pMDI exhibited decreasing tensile strength with increasing WF content. This is a typical phenomenon in incompatible composites. The weak interfacial adhesion between the WF and PBS resulted in a decrease in the tensile strength [4]. In contrast, the tensile strength of the composite with pMDI (2 phr) increased from 40.2 to 49.3 MPa with an increase in WF content from 10 to 50 wt %. In the PBS/WF composites containing pMDI, urethane linkages can be formed during twin-screw extrusion at high temperatures through the reaction of the –NCO groups of the pMDI with hydroxyl groups in the PBS and WF. This urethane linkage may result in the observed improvement in the tensile strength. Neat PBS has a low elastic modulus of 287 MPa, while WF usually has a high elastic modulus of approximately 10–20 GPa. Thus, the elastic modulus of the PBS/WF composites was increased with increasing WF content, regardless of the addition of pMDI, resulting in a decrease in the elongation at break. At the same WF content, the composites with 2 phr pMDI have a slightly higher elastic modulus than those without pMDI. This phenomenon may occur because the increase in the interfacial adhesion caused by the addition of pMDI reduces the voids in the composites.

The effect of KL addition on the PBS/WF (50/50) composites is shown in Figure 5. The tensile strength of the composites decreased with increasing KL content in samples both with and without pMDI. Because KL is an amorphous polymer that has a low strength [32], the strength of the composite with KL decreased significantly with increasing KL content. In a previous study [13], the decrease in tensile strength of the KL/PCL composites due to the increase in KL content was overcome by adding pMDI to the composites. The tensile strength of the KL/PCL composite was improved by approximately 20% as a result of the addition of 2 phr pMDI. Figure 4 also indicates that the PBS/WF/KL composites with pMDI have higher tensile strengths than the composites without pMDI. This demonstrates that pMDI is highly suitable for improving the interfacial adhesion between the biodegradable aliphatic polymer and the wood component owing to the formation of the urethane linkage. The elastic modulus of the PBS/WF/KL composites increased significantly with increasing KL content owing to the high elastic modulus of KL, which is in the range of 2–3 GPa. As with the PBS/WF composites in Figure 4, the PBS/WF/KL composites with pMDI have higher elastic modulus than the samples without pMDI. The elongation at break gradually decreased with increasing KL content, and was not significantly affected by the addition of pMDI.

### 3.3. Melt Flowability

The effect of the WF and KL content and the addition of pMDI on the MFI and MV of the PBS/WF and PBS/WF(50/50)/KL composites is shown in Figure 6. As the WF content increased, the MFI and MV of the PBS/WF composites increased and decreased, respectively. The values of the MFI and MV were inversely proportional. Because WF cannot be melted, the addition of WF deteriorates the melt flow properties of the composites, causing an increase in the MV. KL was added to the PBS/WF (50/50) composites, and the MFI and MV of the PBS/WF(50/50)/KL composites were investigated with varying KL contents. As the content of lignin increased to 20 phr, the MFI increased and the MV decreased. The glass transition temperature (*T*_g_) of KL is between 90 and 150 °C, depending on the wood species, molecular weight, and molecular structure. In addition, KL has a softening point in the range of 130–180 °C, at which the *T*_g_ of KL can occur. These KL thermal behaviors function to increase the MFI of the PBS/WF/KL composites, accompanying the decrease in the MV. The addition of pMDI results in an increase in the MFI of the PBS/WF and PBS/WF(50/50)/KL composites showing a decrease in the MV. This may be due to the urethane linkages connecting the PBS and the WF and KL, which worsen the melt flowability of the PBS matrix. 

The MV of the PBS/WF (50/50) and PBS/WF/KL (50/50/20) composites with and without pMDI was measured at different shear rates by increasing the load from 2.16 kg to 5.0, 7.5, 10.0, and 12.5 kg, as shown in Figure 7. The PBS/WF/KL (50/50/20) sample had a lower MV than the PBS/WF (50/50) sample at a similar shear rate owing to the presence of the lignin. In addition, the PBS/WF/KL (50/50/20) samples exhibited viscosities in a greater range of shear rates. As discussed above, the addition of pMDI caused deterioration of the melt flowability, resulting in a reduction in the MV and the range of shear rates. In all samples, as the shear rate increased, MV decreased gradually. This decreasing trend in the MV exhibits shear thinning behavior.

### 3.4. Thermal Properties

Figure 8 shows the TGA and DTG curves for neat PBS, PBS/WF (50/50) composite, and PBS/WF(50/50)/KL composite. The neat PBS began to exhibit thermal degradation at 300 °C, which accelerated rapidly in the range of 350–430 °C. At the WF sample, the weight loss below 100 °C is related to the removal of the water in the WF. In the DTG curves of the WF, left shoulder peak at about 310 °C is related to the thermal degradation of the cellulose, and main peak corresponds to the degradation of the cellulose at about 360 °C. The PBS/WF and the PBS/WF/KL composites show similar trends in thermal degradation with increasing temperature. The thermal degradation of these composites starts at 230 °C, and continues to 400–420 °C, indicating a lower initial temperature of thermal degradation than that of the neat PBS. In the DTG curves, the PBS/WF and the PBS/WF/KL composites exhibit two peaks at 355 and 390 °C, corresponding to the left shoulder peak and the main peak, respectively, whereas the neat PBS has one main peak at 400 °C. The left shoulder peak at 355 °C may correspond to the decomposition of the hemicellulose and cellulose in the WF. The main peak, with a maximum point at 390 °C, was attributed to the thermal degradation of the neat PBS. The thermal degradation of lignin usually occurs over a wide temperature range of 200–800 °C and is not significant with increasing temperature compared to the degradation of the cellulose, hemicellulose, and PBS. Thus, increasing the KL content caused a decrease in the height of the main peak corresponding to the decomposition of PBS and cellulose in the DTG. The PBS/WF/KL (50/50/20) composite with pMDI (2 phr) exhibited a lower height of the main peak in the DTG curve than the sample without pMDI. This result demonstrates that the addition of pMDI slightly improved the thermal resistance of the composite. 

The non-isothermal melt crystallization behavior of the neat PBS, PBS/WF (50/50) composite, and the PBS/WF/KL (50/50/20) composite with and without pMDI was investigated from the molten samples at cooling rates of 10, 20, 30, and 40 °C. Figure 9 shows the exotherms and relative degree of crystallinity, *X(T)*, of the composites as a function of the temperature. The exotherms of the PBS/WF (50/50) composite occurred at a higher temperature than those of the neat PBS, whereas the exotherms of the PBS/WF/KL (50/50/20) composites with and without pMDI were observed at a lower temperature than those of the neat PBS and the PBS/WF (50/50) composite. As the cooling rate increased, the exotherms became broader and shifted to lower temperatures. The values of the melt crystallization temperature (*T_c_*), crystallization enthalpy (Δ*H_c_*), and crystallinity of the PBS matrix (*X_PBS_*) under non-isothermal conditions are summarized in Table 2. 

The *T_c_* values for the PBS/WF (50/50) composites were higher than those for the neat PBS at the same cooling rate. The addition of WF accelerates the nucleus formation in the molten PBS. However, the *T_c_* values of the PBS/WF/KL (50/50/20) composites with and without pMDI were lower than that of the neat PBS, indicating a delayed nucleation of PBS crystallization with the addition of lignin. The values of Δ*H_c_* were calculated by integrating the crystallization peaks. The *X_PBS_* values for the composites were determined by comparing the Δ*H_c_* values based on the crystallization enthalpy of 100% crystalline PBS, which was assumed to be 210 J/g, and by considering the composition of the PBS in the composites. The *X_PBS_* values of the neat PBS were in the range of 30.7%–32.9% with different cooling rates. The PBS/WF (50/50) composite had *X_PBS_* values of 37.2%–38.2%, which are higher than those of the neat PBS. The addition of 50 wt % WF not only accelerated the nucleation of the PBS, but also improved the *X_PBS_* values. With the addition of KL to the PBS/WF composites, the *X_PBS_* values decreased to 31.2%–32.3% depending on the cooling rate, regardless of the addition of pMDI. 

The relative degree of crystallinity dependent on the temperature, *X(T),* was calculated with Equation (1) as follows:(1)$$\int_{T_{o}}^{T}\left(\frac{dH}{dT}\right)dT/\int_{T_{o}}^{T_{\infty }}\left(\frac{dH}{dT}\right)dT,$$
where *T_o_* and *T*_∞_ are the onset and end crystallization temperatures, respectively; and Δ*H* is the enthalpy of crystallization in an infinitesimal temperature range (*dT*)dT).

. As the cooling rate increased, the range in which the crystallization occurred also increased and shifted to a lower temperature. The values of *X*(*T*) dependent on the temperature can be transformed into the relative degree of crystallinity dependent on time, *X*(*t*), by changing the temperature range to a time scale using Equation (2):(2)$$t=\left(T_{o}-T\right)/\gamma ,$$
where *T_o_* is the onset temperature, *T* is the temperature at crystallization time (*t*), and γ is the cooling rate (°C/min). *X*(*t*) can then be written as Equation (3):(3)$$\int_{T_{o}}^{T}\left(\frac{dH}{dt}\right)dt/\int_{T_{o}}^{T_{\infty }}\left(\frac{dH}{dt}\right)dt,$$

The non-isothermal crystallization kinetics were examined using the Avrami Equation (4):(4)$$1-X\left(t\right)=exp\left(-kt^{n}\right),$$
where *t* is the time, *k* is the crystallization rate constant, and *n* is the Avrami exponent. The Avrami equation can be rewritten as follows:(5)log[−ln(1−X(t))]=logk+nlogt,

The Avrami plots of log[−ln(1−X(t))] versus logt with different cooling rates are shown in Figure 10. The values of *n* and *k* were calculated from the slope and intercept, respectively, of the Avrami plots in Figure 10. The *k* for non-isothermal crystallization can be adjusted using the following equation:(6)$$\log K_{non}=(\log k)/\gamma ,$$
where $K_{non}$ is the crystallization rate constant for non-isothermal crystallization.

The values of the half-crystallization time ($t_{1/2}$), exponent *n*, and $K_{non}$ are summarized in Table 2. The values of $t_{1/2}$ decreased with increasing cooling rate in all samples. With the addition of 50 wt % WF to the PBS polymer, $t_{1/2}$ decreased significantly, indicating that the crystallization became faster with the addition of WF. However, the PBS/WF/KL composites with and without pMDI had similar $t_{1/2}$ values as the neat PBS. It is supposed that the addition of KL may reduce the rate of crystallization. The exponent, *n*, indicates the type of nucleation and dimensionality of the crystal growth. The *n* values of the neat PBS and the PBS/WF (50/50) composite varied from 2.9 to 3.0 and from 2.6–2.9, respectively, indicating that the crystallization proceeds as three-dimensional spherical growth from instantaneous nuclei. The values of *n* for the PBS/WF/KL (50/50/20) composites without pMDI range from 3.2 to 3.5, while the composites with pMDI had *n* in the range of 3.5–4.0, which is slightly higher than for the composites without pMDI. These results may indicate the complex mixture of homogeneous and heterogeneous crystallization. It is suggested that the addition of KL affected the mechanism of nucleation and crystal growth of the PBS matrix. The PBS/WF (50/50) composites had slightly higher $K_{non}$ values, indicating faster three-dimensional crystallization than the neat PBS and the PBS/WF/KL composites with and without pMDI.

## 4. Conclusions

The effect of the addition of KL and pMDI on the morphological characteristics, tensile properties, melt flowability, and thermal properties of PBS/WF composites was investigated. With increasing WF and KL content, the fractured surface of the composites without pMDI became rougher, while pMDI addition improved the interfacial adhesion, resulting in a smooth fractured surface. Increasing the WF and KL content led to a deterioration of the tensile properties of the composites without pMDI; however, pMDI addition improved the tensile properties. The MFI and MV of the PBS/WF composites decreased and increased, respectively, with increasing WF content. The addition of KL increased the MFI, decreasing the MV of the composites, whereas pMDI addition decreased the MFI. The thermal degradation temperatures of the PBS/WF and PBS/WF/KL composites were lower than that of the neat PBS owing to the weaker thermal resistance of WF and KL than of the neat PBS. The addition of WF, KL, and pMDI affected the nucleation and crystallization behavior of the PBS.

## Figures and Tables

**Figure 1 polymers-11-01161-f001:**
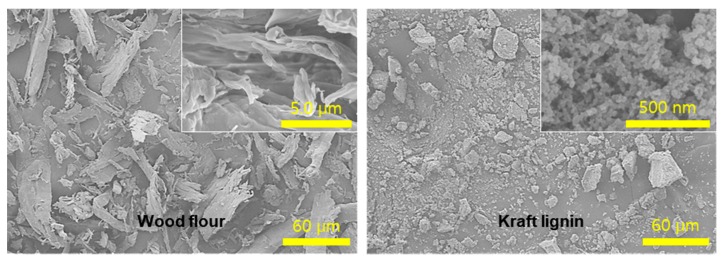
Scanning electron microscope (SEM) micrographs of the wood flour (WF) and kraft lignin (KL).

**Figure 2 polymers-11-01161-f002:**
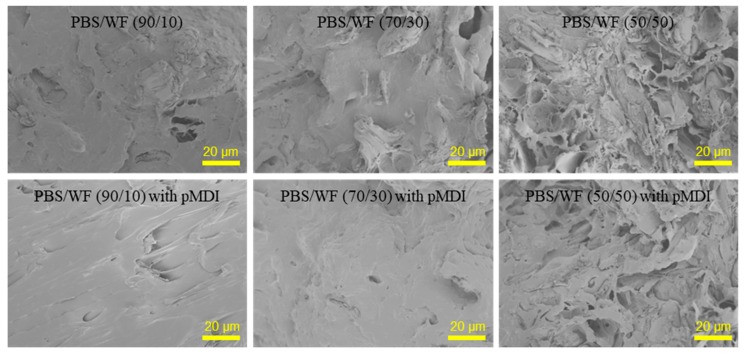
SEM micrographs of the fractured surface of poly(butylene succinate) (PBS)/WF composites with and without polymeric diphenylmethane diisocyante (pMDI) (2 phr).

**Figure 3 polymers-11-01161-f003:**
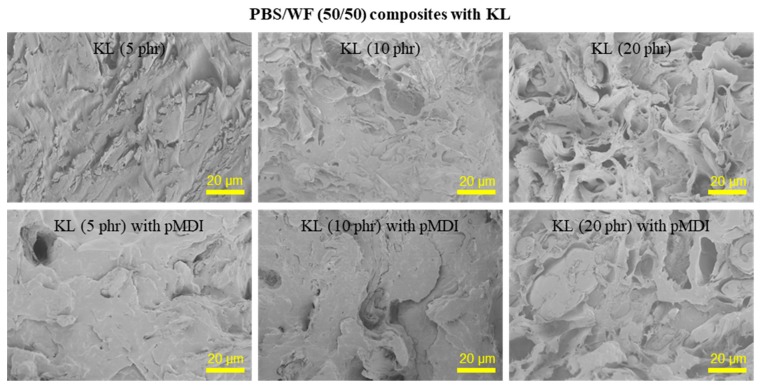
SEM micrographs of the fractured surface of PBS/WF (50/50) composites with varying KL contents with and without pMDI (2 phr).

**Figure 4 polymers-11-01161-f004:**
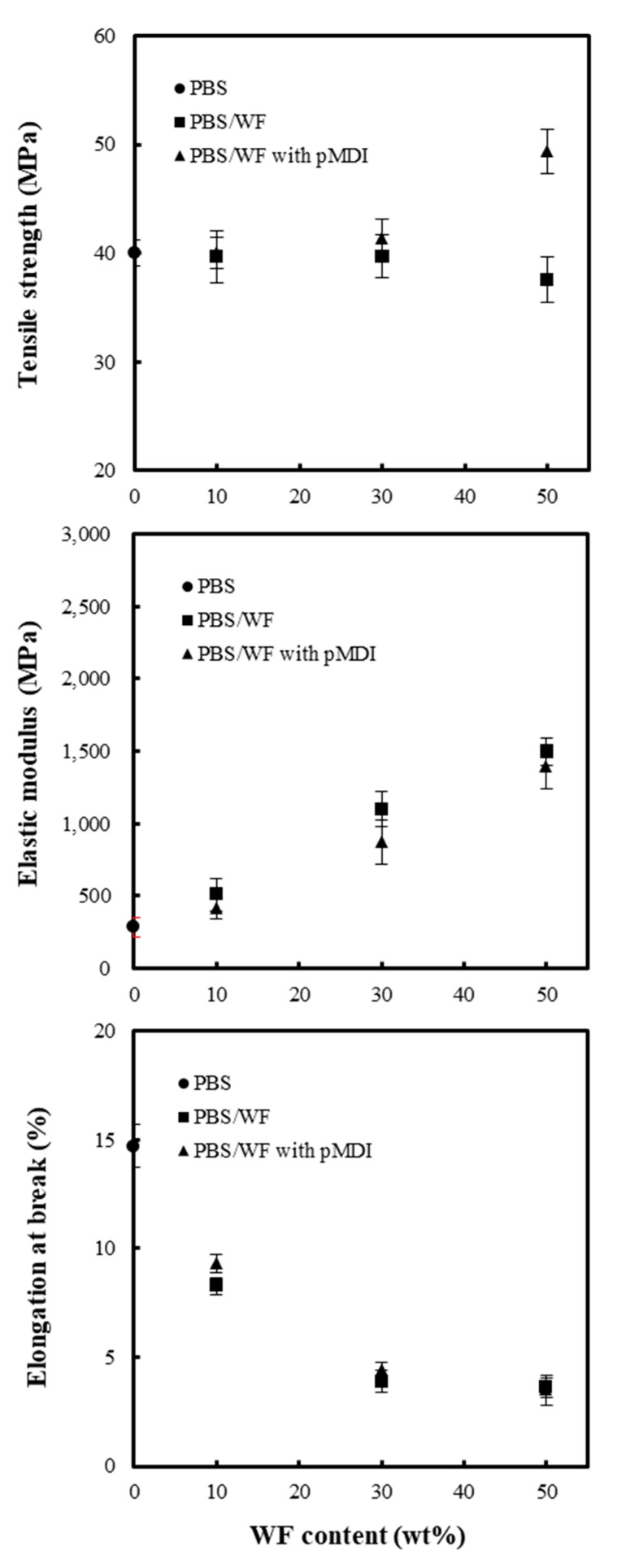
Tensile strength, elastic modulus, and elongation at break of PBS/WF composites with and without pMDI (2phr).

**Figure 5 polymers-11-01161-f005:**
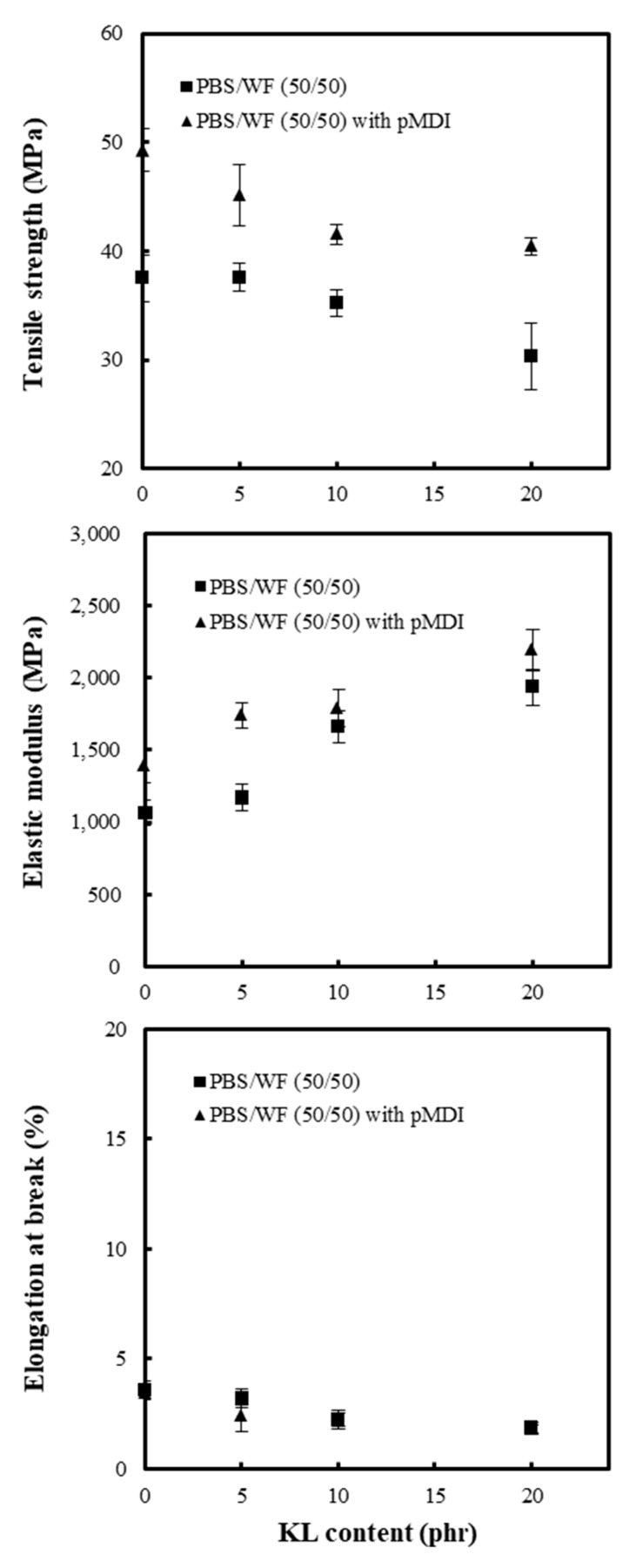
Tensile strength, elastic modulus, and elongation at break of PBS/WF/KL composites with and without pMDI (2phr).

**Figure 6 polymers-11-01161-f006:**
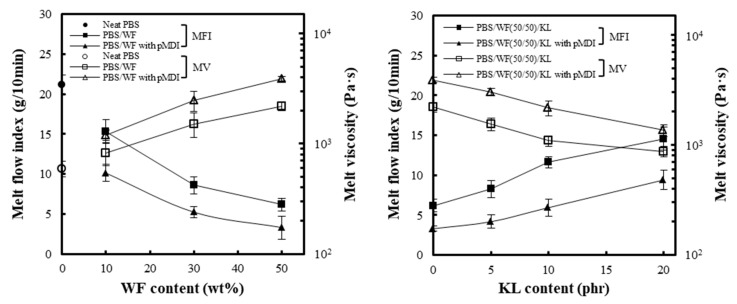
Melt flow index (MFI) and melt viscosity (MV) of PBS/WF composites with varying WF and KL contents.

**Figure 7 polymers-11-01161-f007:**
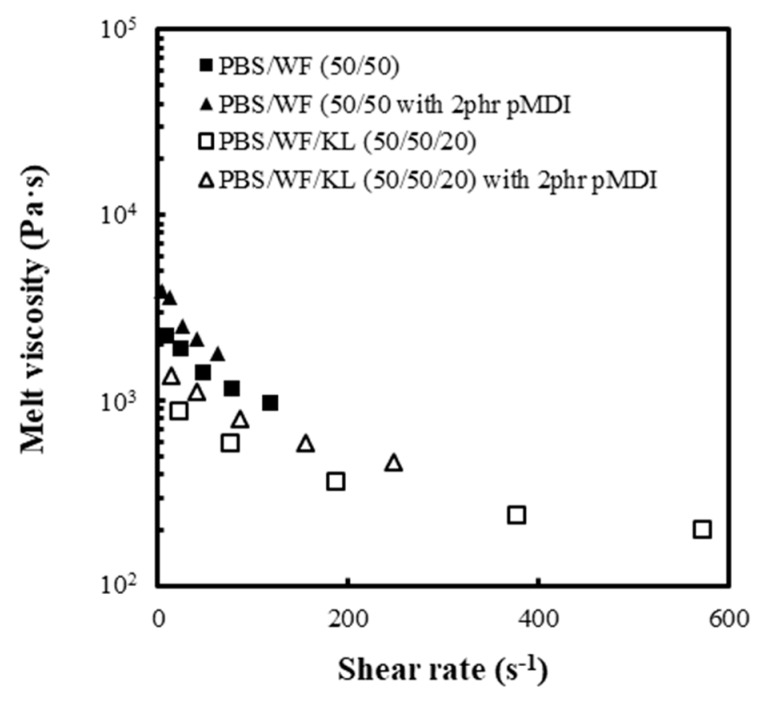
Dependency of the MV of PBS/WF and PBS/WF/KL composites with and without pMDI on the shear rate.

**Figure 8 polymers-11-01161-f008:**
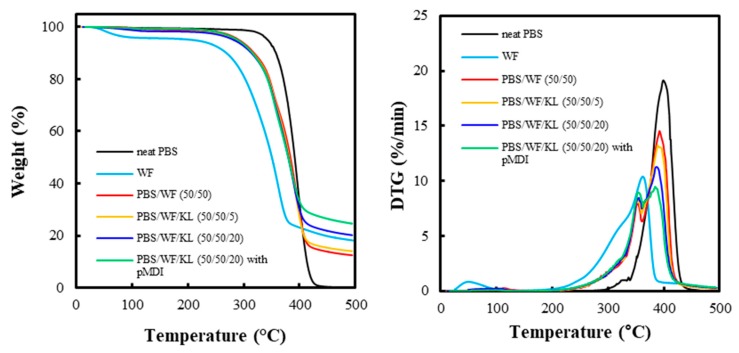
Thermogravimetric analysis (TGA) and derivative thermogravimetry (DTG) curves for the PBS/WF and PBS/WF/KL composites with and without pMDI.

**Figure 9 polymers-11-01161-f009:**
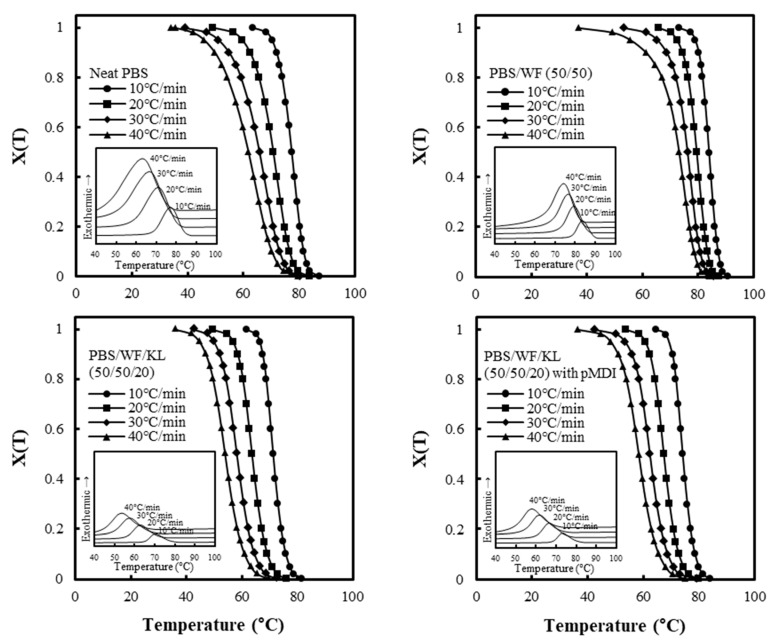
Crystallization exotherms and plots of the relative degree of crystallinity, X(T), versus the temperature for neat PBS, PBS/WF (50/50), and PBS/WF/KL (50/50/20) with and without pMDI at different cooling rates.

**Figure 10 polymers-11-01161-f010:**
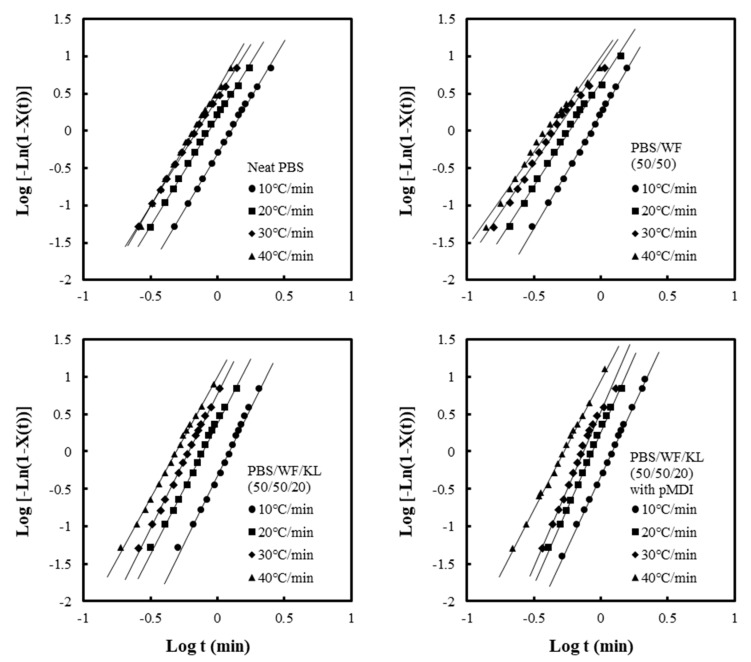
Avrami plots of log[1 − ln(1 − X(t))] versus log *t* for neat PBS, PBS/WF (50/50), and PBS/WF/KL (50/50/20) with and without pMDI at different cooling rates.

**Table 1 polymers-11-01161-t001:** Composite material formulation of the wood-plastic composite (WPC) in weight percentage.

Sample Code	Composition of WPC			
	PBS (%)	WF (%)	KL (phr)	pMDI (phr)
PBS	100	-	-	-
PBS/WF (90/10)	90	10	-	-
PBS/WF (70/30)	70	30	-	-
PBS/WF (50/50)	50	50	-	-
PBS/WF/KL (50/50/5)	50	50	5	-
PBS/WF/KL (50/50/10)	50	50	10	-
PBS/WF/KL (50/50/20)	50	50	20	-
PBS/WF (90/10) with pMDI	90	10	-	2
PBS/WF (70/30) with pMDI	70	30	-	2
PBS/WF (50/50) with pMDI	50	50	-	2
PBS/WF/KL (50/50/5) with pMDI	50	50	5	2
PBS/WF/KL (50/50/10) with pMDI	50	50	10	2
PBS/WF/KL (50/50/20) with pMDI	50	50	20	2

**Table 2 polymers-11-01161-t002:** Crystallization temperature (*Tc*), crystallization enthalpy (Δ*H_c_*), crystallinity of the PBS (*X_PBS_*), half-crystallization time $\left(t_{1/2}\right)$, Avrami exponent (*n*), and corrected crystallization rate constant (Δ$k_{non}$ ) for the neat PBS, PBS/WF (50/50).

Sample Code	γ (°C/min)	*T_c_* (°C)	Δ*H_c_* (J/g)	*X_PBS_* (%)	*t*_1/2_ (min)	*n*	*R* ^2^	Δ*k_non_* (min^−*n*^)
Neat PBS	10	77.1	69.0	32.9	1.12	3.0	0.998	0.93
	20	71.0	66.9	31.9	0.75	2.9	0.998	1.02
	30	66.6	65.2	31.0	0.61	2.9	0.999	1.03
	40	63.1	64.5	30.7	0.58	3.1	0.998	1.03
PBS/WF (50/50)	10	83.6	39.9	38.0	0.77	2.9	0.998	1.05
	20	79.4	39.3	37.4	0.50	2.8	0.994	1.08
	30	76.6	39.0	37.2	0.39	2.7	0.990	1.11
	40	74.2	40.1	38.2	0.34	2.6	0.989	1.12
PBS/WF/KL (50/50/20)	10	70.5	28.1	32.1	1.10	3.5	0.994	0.93
	20	63.0	27.9	31.9	0.71	3.4	0.996	1.04
	30	57.7	27.4	31.3	0.56	3.5	0.998	1.06
	40	53.5	27.3	31.2	0.45	3.2	0.998	1.06
PBS/WF/KL (50/50/20) with pMDI	10	73.3	27.8	32.3	1.13	3.8	0.998	0.93
	20	66.6	27.5	32.0	0.79	4.0	0.997	1.03
	30	61.8	27.2	31.6	0.67	4.0	0.989	1.04
	40	58.9	27.6	32.1	0.47	3.5	0.998	1.06

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
