# Peer review of "Influence of Lignin and Polymeric Diphenylmethane Diisocyante Addition on the Properties of Poly(butylene succinate)/Wood Flour Composite"

_polymers, 2019, doi:10.3390/polym11071161_

Round 1

Reviewer 1 Report

This paper deals with the addition effects of Kraft lignin (KL) and pMDI with poly(butylene succinate) (PBS)/wood flour (WF) composites. These composites with biomass ingredients are effective for reduce of fossil plastics. However, purpose and reason of the KL addition were not clear because WF (including lignin and cellulose) was already added to PBS without preferable properties’ improvement.

Revision points for major are as follows;

1. Purpose and reason of KL addition should be clearly explained in Introduction section.

2. phr (parts by weight per 100 parts by weight of rubber) unit is not general for polymer composites. Ingredients amounts of composites should be explained as mass% of total composite weight. However, these revisions are almost impossible at this moment. So, compositions of all polymer composites (formulation) used in this paper should be explained in Table.

3. Only WF should be measured by TGA. (Figure 7)

Revision points for minor are as follows;

1. pMDI in title should be “polymeric diphenylmethane diisocyanate”.

2. pMDI at first appearance in Abstract should be “polymeric diphenylmethane diisocyanate (pMDI)”.

3. KL in abstract should be “Kraft lignin (KL)”.

4. Average size, shape, aspect ratio, composition, resource and so on of WF and KL should be explained in Materials and Methods.

5. Making procedure of WF and KL should be explained simply.

6. (Line 155) “hydroxy groups of PBS” means terminal only one hydroxy group in one PBS polymer chain. Is this crosslinking point structure right?

7. (Line 169) “KL is amorphous polymer”. What’s the evidence data?

8. (Line 173) “3 phr” is not correspondence to Figure 4 (phr 2).

9. (Figure 7) Why was final weight of PBS only by TGA, not zero?

Author Response

Revision points for major are as follows;

Comment 1.
Purpose and reason of KL addition should be clearly explained in Introduction section.

Response 1.
Thank you for your comment. In general, as the wood flour content increases in wood/plastic composites(WPC), the melt flowability of the composite decreases. This phenomenon results in deterioration of the workability and quality of the WPC. In order to overcome this problem, we added the lignin which is beneficial to improve the flowability. We added the explanation for the reason of the lignin addition in the manuscript (line 54-56).

Comment 2.

phr (parts by weight per 100 parts by weight of rubber) unit is not general for polymer composites. Ingredients amounts of composites should be explained as mass% of total composite weight. However, these revisions are almost impossible at this moment. So, compositions of all polymer composites (formulation) used in this paper should be explained in Table.

Response 2.

Thank you for your valuable comment. We’ve thought that the unit “phr” (per parts hundred rubber/resin) is often used in the composite science. The some latest articles on the composites in Polymers journal have also used the unit “phr”, as like below;

1. Hou et al. Super-toughned poly(lactic acid) with poly(ε-caprolactone) and ethylene-methyl acrylate-glycidyl methacrylate by reactive melt blending polymers. Polymers 2019, 22(5), 771.

2. Xia et al. Shape memory behavior of carbon black-reinforced trans-1,4-polyisoprene and low-density polyethylene composites. polymers 2019, 11(5), 807.).

However, we’ve thought that we need to explain the composition of the composite in detail, so we’ve inserted the Table 1, indicating material formulation of each sample in the manuscript for readers’s better understanding.

Thank you.

Comment 3.

Only WF should be measured by TGA. (Figure 7)

Response 3.

Thank you for your valuable review. TGA data for only WF was added in Figure 7. We’ve expected that the result may help to understand the effect of WF content on the thermal properties of the WPC composites. Thank you.

Revision points for minor are as follows;

Comment 4.

pMDI in title should be “polymeric diphenylmethane diisocyanate”.

Response 4.

We’ve added the full name of ‘pMDI’ as like ‘polymeric diphenylmethane diisocyante’ in the title. Thank you.

Comment 5. pMDI at first appearance in Abstract should be “polymeric diphenylmethane diisocyanate (pMDI)”.

Response 5.

In the abstract, we changed the abbreviation ‘pMDI’ to ”polymeric diphenylmethane diisocyanate”. Thank you.

Comment 6.

KL in abstract should be “Kraft lignin (KL)”.

Response 6.

Thank you for your comment. We modified the word ‘KL’ to ‘kraft lignin’ in the abstract.

Comment 7.

Average size, shape, aspect ratio, composition, resource and so on of WF and KL should be explained in Materials and Methods.

Response 7.

Thank you for your profitable comment. We’ve agreed that the characteristics of the WF and KL should be explained in the manuscript. Therefore, we’ve observed the morphological characteristics of the WF and KL by using the scanning electron microscope. SEM micrographs were added in Figure 1. and the morphological characteristics were discussed in the manuscript. Thank you.

Comment 8.

Making procedure of WF and KL should be explained simply.

Response 8.

Thank you for your comments. We purchased the WF and used it in this research. We already inserted the information on the commercial WF and stated the making procedure of the KL in the manuscript. Thank you very much.

Comment 9.

(Line 155) “hydroxy groups of PBS” means terminal only one hydroxy group in one PBS polymer chain. Is this crosslinking point structure right?

Response 9.

The hydroxyl groups in the PBS exist at the terminal end of the PBS. As a result of pMDI addition, the urethane linkage can occur between the hydroxyl groups of the PBS and aliphatic or phenolic hydroxyl groups of the KL. Thank you.

Comment 10.

(Line 169) “KL is amorphous polymer”. What’s the evidence data?

Response 10.

Thank you for your comment. We added the reference in the sentence “KL is amorphous polymer”. It may help to make it clear that the KL is amorphous.

Comment 11.

(Line 173) “3 phr” is not correspondence to Figure 4 (phr 2).

Response 11

Thank you. It was a typing error. We modified the ‘3 phr’ to the ‘2phr’. 

Comment 12.

(Figure 7) Why was final weight of PBS only by TGA, not zero?

Response 12.

Thank you for your review. We’ve agreed on your comment. It had some mistakes during analyzing the data. We modified it correctly. Thank you again. 

Reviewer 2 Report

The paper by Park et al  is focused on the study of PBS modified with wood  flour. The approach proposed is not entirely novel and it has been studied in the past. However, the paper is still interesting for those involved in the field. The paper is clearly written and assembled. However, it needs some major review to address better some of the experimental results reported before it can accepted.          

Abstract line 17-18. When the authors report KL for the first time they should explain what they refer to. Kraft Lignin I assume.

KL was obtained by the authors as they explained in the experimental part. However, no chemical characterization of the KL is reported here. This should be added. In addition to that the wood flour particles should be characterized by SEM.

The authors at pag.3 Line 109-113 discuss the morphological features of their composites. The differences in morphology with the addition of pMDI are not so relevant as claimed by the authors. In addition, if the adhesion was improved some WF particles should be observed. If one looks to the macroscopic mechanical properties (fig.3) the addition of pMDI seems to have a relevant effect for the 50wt% WF formulation only. The authors should review their comments to account for this.

The results on the crystallization kinetic study should be commented referring to the effect on processing too.

Author Response

Comment 1.

Abstract line 17-18. When the authors report KL for the first time they should explain what they refer to. Kraft Lignin I assume.

Response 1.

Thank you for your valuable review. We revised the word ‘KL’ to ‘kraft lignin’.

Comment 2.

KL was obtained by the authors as they explained in the experimental part. However, no chemical characterization of the KL is reported here. This should be added.

Response 2.

Thank you for your review. Our research team has conducted various researches on the lignin chemistry and lignin-based composite. We’ve already released various chemical characteristics of the KL in the previous studies cited in the manuscript. We’ve expected that the references may help readers understand the characteristic of the KL. Thank you very much.

Comment 3.

In addition to that the wood flour particles should be characterized by SEM.

Response 3.

Thank you for your comment. We’ve agreed on your comment. We added the SEM image of the WF and discussed the morphologies in the manuscript. Thank you.

Comment 4.

The authors at pag.3 Line 109-113 discuss the morphological features of their composites. The differences in morphology with the addition of pMDI are not so relevant as claimed by the authors. In addition, if the adhesion was improved some WF particles should be observed. If one looks to the macroscopic mechanical properties (fig.3) the addition of pMDI seems to have a relevant effect for the 50wt% WF formulation only. The authors should review their comments to account for this.

Response 4. Thanks for critical point. When WF content increase, the interfacial region (area) will be increased. Therefore, the difference of mechanical strength between the composites with and without pMDI will become larger in the composites with larger amount of WF, because the interfacial area with the improved adhesion due to pMDI addition will be increased in the composites with larger amount of WF. Thanks.

Comment 5.

The results on the crystallization kinetic study should be commented referring to the effect on processing too.

Response 5.

Thank you for your review. We’ve agreed on your comment. As expected, the addition of the WF increased the rate of crystallization of the PBS and crystallinity index of the PBS. However, as a result of the KL addition, the crystallization rate rather decreased both the rate of crystallization and crystallinity index of the PBS. We are investigating the effect of KL addition on the crystallization kinetic of the PBS-based composites. After that, we will release the results in the new article.

Round 2

Reviewer 1 Report

I confirm the revisions based on m comments.